Bacteria associated with human saliva are major microbial components of Ecuadorian indigenous beers (chicha)

Freire Ana L.
Zapata Sonia
Mosquera Juan
Mejia Maria Lorena
Trueba Gabriel gabrieltrueb36@hotmail.com
Instituto de Microbiologia, Colegio de Ciencias Biologicas y Ambientales, Universidad San Francisco de Quito , Quito, Pichincha , Ecuador
Smidt Hauke
Electronic publication date: 2016 Apr 28
Publication date: 2016
Volume: 4
Electronic Location ID: e1962
Received 2015 Nov 19; Accepted 2016 Apr 1
Copyright: © 2016 Freire et al.
Copyright year: 2016
Copyright holder: Freire et al.
License: This is an open access article distributed under the terms of the Creative Commons Attribution License, which permits unrestricted use, distribution, reproduction and adaptation in any medium and for any purpose provided that it is properly attributed. For attribution, the original author(s), title, publication source (PeerJ) and either DOI or URL of the article must be cited.
License URL: https://creativecommons.org/licenses/by/4.0/

Keywords: Lactic acid bacteria, Indigenous beer, Fermentation, Chicha, Microbiota, Artisanal fermented beverages, Streptococcus salivarius, Streptococcus mutans, Lactic acid bacteria, Fermented cassava, Ecuador, Chewed indigenous beer, Cassava, Saliva

Funding: This project was conducted with funds from CSK food enrichment, Netherlands. The funders had no role in study design, data collection and analysis, decision to publish, or preparation of the manuscript.

==============================
Indigenous beers (chicha) are part of the indigenous culture in Ecuador. The fermentation process of these beers probably relies on microorganisms from fermented substrates, environment and human microbiota. We analyzed the microbiota of artisanal beers (including a type of beer produced after chewing boiled cassava) using bacterial culture and 16S ribosomal RNA (rRNA) gene-based tag-encoded FLX amplicon pyrosequencing (bTEFAP). Surprisingly, we found that Streptococcus salivarius and Streptococcus mutans (part of the human oral microbiota) were among the most abundant bacteria in chewed cassava and in non-chewed cassava beers. We also demonstrated that S. salivarius and S. mutans (isolated from these beers) could proliferate in cassava mush. Lactobacillus sp. was predominantly present in most types of Ecuadorian chicha.

Introduction

The domestication of fermenting bacteria and yeast predated the domestication of animals and plants; ancestral hominids adapted to metabolize alcohol long before the Neolithic period (Carrigan et al., 2015). The organoleptic and psychotropic effects associated with the consumption of accidentally fermented fruits or cereals may have motivated early humans to replicate this process. Additionally, fermentation may have provided unintended benefits as fermenting bacteria may have reduced the risks of foodborne diseases in ancient societies (Nakamura et al., 2012; Lewus, Kaiser & Montville, 1991; Fooks & Gibson, 2002; Tesfaye, Mehari & Ashenafi, 2011); it is still unclear whether these microorganisms confer additional health benefits (McNulty et al., 2011). The use of alcoholic beverages has played a crucial role in the evolution of human societies (Joffe, 1998); nevertheless, very little is known about the process of domestication and evolution of these fermenting microorganisms (Libkind et al., 2011).

Many fermenting microorganisms have originated in the environment and food substrates (Martini, 1993) while others resemble microorganisms found in the human microbiome, suggesting human (skin or intestine) origins (Agapakis & Tolaas, 2012); in fact, some modern fermented dairy products contain intestinal bacteria (Walter, 2008).

Indigenous people from South America (such as in Ecuador) prepare a type of beer known as chicha which is made with either corn, boiled cassava or the fruit of the palm Bactris gasipaes (chonta); some cassava beers include an additional chewing step before the fermentation process. A recent report showed that bacteria present in chewed cassava beers were mainly Lactobacillus sp. (Colehour et al., 2014). We analyzed the microbial diversity (using culture dependent and culture independent techniques) in different types of Ecuadorian chicha.

Materials and Methods

Sample collection

Four samples of chicha (indigenous beer) from two geographical regions of Ecuador (Andean and Amazon regions) were collected. These samples included beer made with both chewed cassava (CC), mushed cassava (MC); mushed chonta (CB) and ground corn (CoB) (Table 1). The samples of CC and MC were purchased from the same household. All these products were obtained from rural communities. None of these beers were pasteurized, nor had they any commercial additives or preservatives. All samples were refrigerated (2–8 °C) after collection; a 2 mL aliquot of sample was stored at −20 °C, for molecular phylotyping.

Table 1 Description and site of collection of the different types of indigenous beers analyzed.

Main ingredient	Substrate scientific name	Geographical region	Site of collection	Time of fermentation	
Chewed cassava	Manihot esculenta	Amazon	Puyo	3 days	
Mushed cassava	Manihot esculenta	Amazon	Puyo	3 days	
Chonta	Bactris gasipaes	Amazon	Tena	2 days	
Corn (jora)	Zea mays	Highlands	Pifo	2 days	

Plate count of lactic acid bacteria (LAB)

A 20 mL aliquot of each sample was homogenized in 180 mL of a sodium citrate solution (10−1 dilution) and ten-fold dilutions were made in saline solution (NaCl 0.9%). One mL of each dilution was inoculated in MRS (pH 5) and M17 (pH 7, 0.5% dextrose) by pour plate method. Two incubation temperatures were used (37 and 43 °C) under aerobic and anaerobic conditions, for 3–5 days. The incubation time varied because of the different bacteria present on each product.

Phenotypic characterization

Ten colonies (showing different morphology) were randomly picked from MRS plates from each sample. A subset of colonies showing characteristics of lactic acid bacteria (oxidase negative, catalase negative, Gram positive rods or cocci) was selected for molecular characterization (Table 2); 5 from CC, 6 from MC, 6 from CB and 8 from CoB. Strains were stored at −20 °C in MRS or M17 broth with 20% of glycerol.

Table 2 Bacteria isolated from the four beer samples.

All the 25 strains were obtained by bacterial cultures in MRS and M17 and 16 S ribosomal gene from colonies was amplified and sequenced.

Sample	Isolate ID	Culture media	Growth condition	Identification (16S ribosomal RNA gene)	
Chewed cassava beer	25 A2	MRS	Anaerobic	Leuconostoc mesenteroides	
25 C2	MRS	Aerobic	Lactobacillus fermentum	
25 E2	M17	Anaerobic	Streptococcus mutans	
25 F1	M17	Aerobic	Lactococcus lactis	
25H1	M17	Aerobic	Streptococcus salivarius	
Mushed cassava beer	26 A1	MRS	Anaerobic	Lactobacillus fermentum	
26 B1	MRS	Anaerobic	Lactobacillus fermentum	
26 C2	MRS	Aerobic	Lactobacillus fermentum	
26 E2	M17	Anaerobic	Streptococcus salivarius	
26 F2	M17	Anaerobic	Streptococcus salivarius	
26 G1	M17	Aerobic	Streptococcus salivarius	
Chonta beer	27 A1	MRS	Anaerobic	Lactobacillus plantarum	
27 B1	MRS	Anaerobic	Weissella confusa	
27 C1	MRS	Aerobic	Weissella confusa	
27 E1	M17	Aerobic	Lactococcus lactis	
27 F2	M17	Anaerobic	Lactococcus lactis	
27 G2	M17	Aerobic	Lactococcus lactis	
Corn beer	61 B2	MRS	Anaerobic	Lactobacillus casei	
61 G1	M17	Anaerobic	Leuconostoc mesenteroides	
61 G2	M17	Anaerobic	Lactobacillus plantarum	
61 H1	MRS	Anaerobic	Lactobacillus parabuchneri	
61 I1	MRS	Anaerobic	Lactobacillus paracasei	
61 J1	MRS	Anaerobic	Lactobacillus pantheris	
61 K1	M17	Anaerobic	Leuconostoc mesenteroides	
61 L1	M17	Anaerobic	Leuconostoc mesenteroides	

Genotypic characterization of bacterial colonies

DNA was extracted from one colony using DNAzol Reagent (Life Technologies, Carlsbad, CA, USA) following manufacturer instructions and the DNA was stored at −20 °C until used. The 16S ribosomal RNA gene was amplified in 25 uL containing: 1X PCR buffer, 2.5 mM MgCl2, 0.25 mM dNTP’s, 0.2 uM 27F primer (5′-AGAGTTTGATCCTGGCTCAG-3′), 0.2 uM 1492R primer (5′-GGTTACCTTGTTACGACTT-3′) (Martin-Laurent et al., 2001), 0.5 U GoTaq Flexi DNA polymerase (Promega, Madison, WI, USA), 5 uL of sample DNA and Milli-Q water. The times and temperatures used for the amplification were: melting (94 °C, 1 min), annealing (56 °C, 30 s), elongation (72 °C, 30 s), this routine was repeated for 30 cycles, and final extension (72 °C, 10 min). Amplicons were subjected to gel electrophoresis (1% agarose gel), sequenced at Functional Biosciences (Madison, WI, USA) and DNA sequences analyzed using Seqmatch (Ribosomal Database Project: http://rdp.cme.msu.edu/) and submitted to GenBank; the accession numbers are KT722809–KT722833).

High throughput sequencing analysis

In order to complement the culture-based protocols, we investigated the microbial diversity using FLX amplicon pyrosequencing. DNA was extracted from all beer samples using DNeasy Plant Mini kit (Qiagen) following manufacturer’s protocols, but instead of using AE buffer for elution, we used same volume of PCR Milli-Q water. DNA samples from four types of beer were sent to CD Genomics (Shirley, NY, USA), for 16S-based phylotyping. DNA was subjected to bacterial tag-encoded FLX amplicon pyrosequencing (bTEFAP) using primers 939F-5′TTGACGGGGGCCCGCAC3′ and 1492R-5′TACCTTGTTACGACTT3′. For fungal sequences we used ITSF-5′CTTGGTCATTTAGAGGAAGTAA3′. Resulting sequences (minimum length = 250 nucleotides) were trimmed and quality scored using USearch (http://drive5.com/); chimeras were detected using UCHIIME (http://drive5.com/) in de novo mode and were compared using BLASTn to a ribosomal database. Identity values were used to make assignments to the appropriate taxonomic levels: greater than 97% identity were resolved at the species level and between 95 and 97% at the genus level. The number of bacterial sequences we obtained were: 2,965 readings for CC, 3,320 for MC, 3,046 for B and 15,623 for CoB. For fungi we obtained 6,763 readings from CC, 6,925 from MC and 6,558 from CB. We did not carry out fungi analysis of CoB. All sequences were submitted to Sequence Read Archive and accession numbers are: SRP070493, SRS1299611, SRX1612367, SRR3202831, SRS1299612, SRX1612366, SRR3202830, SRS1299613, SRX1612365, SRR3202829, SRS1310202, SRX1600290, SRR3187397, SRS1310203, SRX1600289, SRR3187396, SRS1310204, SRX1600288, SRR3187395, SRS1310207, SRX1612364, SRR3202828, SRS1310208, SRX1600292, and SRR3202832.

Streptococcus salivarius and Streptococcus mutans growth in cassava solution

To rule out the possibility of S. salivarius or S. mutans contamination, one colony of a pure culture of each bacteria (obtained from beers) was diluted in 25 mL of sodium citrate (2%) separately. Subsequently, 1 mL of this cell suspension was used to inoculate tubes containing 9 mL of sterile (autoclaved) chewed cassava solution (10%) and incubated at 37 °C under anaerobic conditions. A 100 μL aliquot from each incubated tube was extracted and plated in M17 (this was done by triplicate) at 0, 24, 48 and 72 h of inoculation. Results from each day were compared to determine the ability of these bacteria to grow in chewed cassava solution.

Statistical analysis

We used Mann-Whitney U test to test whether S. salivarius and S. mutans were able to grow in cassava solution. Shannon indices were calculated using the formula H = −Σpilog(pi), pi being the relative frequency of the abundance of each species found. Principal component analysis (PCA) of the bacterial species and abundance of the four beverages was performed using the software SPSS v21 (IBM Corp, Armonk, NY, USA).

Results

Characterization of bacterial isolates

Twenty-five bacterial isolates (cultured from the four beer types) were characterized by 16S rDNA sequencing showing 99–100% identity when compared with GenBank sequences (Table 2). The predominant bacterial species in all beers were Lactobacillus fermentum (16%), Lactococcus lactis (16%), Leuconostoc mesenteroides (16%), and Streptococcus salivarius (16%); followed by Lactobacillus plantarum (8%), Weissella confusa (8%), Lactobacillus casei (4%), Lactobacillus pantheris (4%), Lactobacillus parabuchneri (4%), Lactobacillus paracasei (4%) and Streptococcus mutans (4%). The most diverse bacterial composition (using culture-dependent techniques) found in CoB (6 bacterial species), followed by the CC (5 bacterial species), CB (3 bacterial species) and MC (2 bacterial species). Intriguingly, cassava beers contained human salivary bacteria: both CC and MC had Streptococcus salivarius while CC had also S. mutans (Table 2).

High throughput sequencing analysis

The beer with greater diversity was CC (31 bacterial species), followed by CoB (26 bacterial species), CB (21 bacterial species), MC (20 bacterial species). The predominant bacterial species in CC were Lactobacillus spp. (40.9%) followed by human microbiota bacteria: Streptococcus salivarius (31.94%), Streptococcus parasanguinis (5.41%), Streptococcus pneumoniae (3.65%). The most prevalent bacteria in MC were Streptococcus spp. (83%) followed by Lactococcus sp. (9.32%); the majority of streptococci have been described as part of the human microbiota: Streptococcus salivarius (65%), Streptococcus pasteurianus (7.74%), and Streptococcus parasanguinis (3.47%). The most prevalent bacteria in CB were Weissella confusa (46%), Weissella sp. (20%), and Lactococcus lactis (9%). The dominant bacteria in CoB were Weissella sp. (19%) and Lactobacillus plantarum (12.5%), Lactococcus garviae (2.76%) Lactobacillus brevis (2.5 %) (Table 3). The dominant fungal species present in different beers analyzed was very similar; Saccharomyces cerevisiae was the most abundant comprising 92% of all the taxa detected (Table 4).

Table 3 Most predominant bacterial species (abundance of more than 0.1%) found by pyrosequencing analysis of samples from 4 types of chicha.

Bacterial species	CC	MC	CB	CoB	Cultured	Possible origins	
Bacillus amyloliquefaciens	0.0	0.5	0.0	0.00	–	Environment	
Carnobacterium maltaromaticum	0.0	0.0	1.0	0.1	–	Environment	
Enterobacter asburiae	0.5	0.0	0.0	0.0	–	Environment	
Enterobacter cancerogenus	0.5	0.0	0.0	0.0	–	Environment	
Enterobacter sp	1.3	0.0	0.1	0.0	–	Environment	
Fructobacillus sp	0.0	0.0	0.0	3.8	–	Vegetables	
Gluconacetobacter intermedius	0.0	0.0	0.0	0.6	–	Fermented food	
Kluyvera ascorbate	0.4	0.0	0.0	0.0	–	Human gut, food	
Lactobacillus brevis	8.4	0.1	2.5	0.6	–	Environment, gut	
Lactobacillus camelliae	0.0	0.0	0.0	7.3	–	Environment, gut	
Lactobacillus casei	0.0	0.0	0.0	3.1	+	Environment, gut	
Lactobacillus delbrueckii	8.0	0.0	0.0	0.0	–	Environment, gut	
Lactobacillus fermentum	6.5	3.8	0.0	0.0	+	Environment, gut	
Lactobacillus harbinensis	0.0	0.0	0.0	2.1	–	Vegetables	
Lactobacillus manihotivorans	1.8	0.0	0.0	0.0	–	Vegetables	
Lactobacillus parabuchneri	0.0	0.0	0.0	1.4	+	Oral microbiota	
Lactobacillus paracasei	0.0	0.0	0.0	8.6	+	Environment, gut	
Lactobacillus paracollinoides	0.0	0.0	0.0	16.0	–	Environment, gut	
Lactobacillus plantarum	10.8	0.0	12.4	0.1	+	Environment, gut	
Lactobacillus sp	3.4	0.0	0.7	1.3	–	Environment, gut	
Lactobacillus vaccinostercus	1.2	0.0	0.2	0.0	–	Environment, gut	
Lactococcus garviae	0.0	0.0	2.8	0.0	–	Fermented food	
Lactococcus lactis	2.1	0.0	8.9	0.0	+	Environment, gut	
Lactococcus sp	0.2	9.3	1.0	0.2	–	Gut	
Leuconostoc citreum	0.0	1.5	1.2	0.0	–	Fermented food	
Leuconostoc lactis	1.7	0.1	0.2	0.8	–	Environment	
Leuconostoc sp	0.0	0.0	0.1	4.6	–	Vegetables	
Oenococcus kitaharae	0.0	0.0	0.0	1.2	–	Vegetables	
Serratia sp	1.0	0.0	0.0	0.0	–	Environment	
Streptococcus gallolyticus	0.0	0.5	0.0	0.0	–	Oral microbiota	
Streptococcus oralis	1.4	0.2	0.0	0.0	–	Oral microbiota	
Streptococcus parasanguinis	5.4	3.5	0.0	0.0	–	Oral microbiota	
Streptococcus pasteurianus	0.0	7.7	0.0	0.0	–	Human gut	
Streptococcus pneumoniae	3.6	0.5	0.0	0.0	–	Human nasopharynx	
S. pseudopneumoniae	0.5	0.0	0.0	0.0	–	Human nasopharynx	
Streptococcus salivarius	32.0	65.0	0.0	0.0	+	Oral microbiota	
Streptococcus sp	2.5	2.3	0.1	0.0	–	Human microbiota	
Streptococcus thermophilus	1.2	2.59	0.0	0.0	–	Vegetables	
Streptococcus vestibularis	0.4	0.8	0.0	0.0	–	Oral microbiota	
Weissella cibaria	0.1	0.0	0.9	0.9	–	Vegetables	
Wbconeissella confusa	0.5	0.1	45.9	25.3	+	Vegetables	
Weissella paramesenteroides	0.5	0.0	0.1	0.0	–	Environment	
Weissella sp	0.2	0.3	19.8	19.4	–	Vegetables	
Note:

Chewed cassava, CC; mushed cassava, MC; chonta, CB; corn, CoB. Numbers indicate percentages and “+” indicates that bacterium recovered in culture.

Table 4 Most predominant fungal species found by pyrosequencing analysis of samples from 3 types of chicha.

Fungal species	CC	MC	CB	Possible origins	
Saccharomyces cerevisiae	92.533	92.023	92.033	Vegetables	
Penicillium citrinum	0.03	0.021	0.062	Soil	
Debaryomyces hansenii	0.636	0.547	0.549	Sea water	
Hanseniaspora uvarum	0.044	0.056	0.075	Vegetables	
Wallemia muriae	0.118	0.115	0.137	Salty water	
Wallemia sp	1.316	1.701	1.602	Salty water	
Aspergillus sp	0.089	0.047	0.032	Soil	
Pichia kudriavzevii	1.05	1.5	1.32	Vegetables	
Aspergillus versicolor	0.104	0.138	0.135	Soil	
Pichia burtonii	0.118	0.123	0.107	Vegetables	
Hyphopichia burtonii	0.089	0.067	0.073	Starch substrates	
Cyberlindnera sp	0.532	0.54	0.545	Waste deposits	
Pichia sp	0.044	0.04	0.054	Soil	
Saccharomyces bayanus	0.104	0.132	0.096	Vegetables	
Galactomyces sp	3.149	2.908	3.133	Rumen, fermented food	
Pichia fermentans	0.044	0.042	0.047	Vegetables	
Note:

Chewed cassava, CC; mushed cassava, MC; chonta, CB. The numbers indicate percentages.

Growth of S. salivarius and S. mutans in cassava solution

Streptococcus salivarius (Fig. 1) and S. mutans (Fig. 2) grew in chewed cassava solution. After 48 h of culture (S. salivarius) and 72 h (S. mutans), the bacterial counts went down.

Figure 1 Growth of S. salivarius in sterile chewed cassava solution.

There is a significant increase in CFU (Mann-Whitney U test) at the 24 h of incubation compared with those at inoculation time (0 h).

Figure 2 Growth of S. mutans in chewed cassava solution.

There is a significate increase in CFU (Mann-Whitney U test) at the 48 h of incubation compared with those at the inoculation time (0 h).

Diversity estimations

CC was the beverage with the most species diversity (H = 1.06, E = 0.71), followed by CoB (H = 0.94, E = 0.66), CB (H = 0.71, E = 0.54), and MC (H = 0.59, E = 0.45). The evenness values followed the same pattern and suggest that CC is also the most heterogeneous in terms of species (Hayek & Buzas, 2010; Pielou, 1966).

Principal component analysis

The type of beer (fermented substrate) accounted for 90.4% of the bacterial species variability and cassava beers had more similar bacterial composition and abundance than the other types of beer; interestingly CB and CoB also showed similarity (Fig. 3).

Figure 3 Principal component analysis of beers’ microbiota.

Beers made with cassava (MC and CC) formed a cluster different from the cluster formed by beers made with either chonta (CB) or corn (CoB). Each pair of beverages that form a group share a similar bacterial species profiles and abundance.

Discussion

Our study found higher bacterial diversity in beer that contained human saliva (Tables 2 and 3); therefore, saliva may not only speed up the fermentation process (by providing amylases as suggested by Henkel (2005)) but may also offer an additional bacterial inoculum which may favor this process. This finding may provide additional explanation for the adoption of such a peculiar process in the beer’s manufacture.

Our study also demonstrated the presence of oral streptococci such as S. salivarius, S. mutans, S. parasanguinis in cassava beers; these bacteria may thrive on carbohydrates present in the oral cavity after starchy meals (Moye, Zeng & Burne, 2014; Burne, 1998). Oral bacteria S. salivarius and S. mutans were cultured from cassava chicha (with saliva and without saliva) in large numbers and were shown to grow in mushed cassava under laboratory conditions. Oral bacteria in beer without human saliva may indicate contamination of fermenting containers (or utensils). Fermenting bacteria are known to produce biofilm in containers (Kebede et al., 2007) and both types of cassava beers were obtained from the same household, and most likely they use the same pots for both type of beers. It is possible that some strains of S. salivarius from these beers may be adapting to the fermentation process; Streptococcus thermophilus, a bacteria used as starter in yogurt (Burton et al., 2006) may have evolved from S. salivarius (Hols et al., 2005). Future studies should investigate the prevalence of S. salivarius in larger number of cassava chichas from other locations and find out whether the strains of S. salivarius isolated from beers are different from those isolated from human saliva.

A recent study failed to detect S. mutans and S. salivarius in chicha prepared with chewed cassava in Ecuador (Colehour et al., 2014). The disagreement between both studies may result from differences in samples in both studies; Colehour et al. (2014) collected beers that were fermenting for four days while we collected samples that were fermenting for three days. Beer microbiota changes overtime (Steinkraus, 2002) and in the case of S. mutans and S. salivarius we observed a sharp increase and decline in bacterial populations in 24 h (Figs. 1 and 2). Unlike Colehour et al. (2014), we also carried out bacterial cultures.

Reduction on streptococci populations may be due to the consumption of all the nutrients, accumulation of toxic metabolites, autolysis (Dufour & Lévesque, 2013). Also, these bacteria are known to form biofilm (Ajdić et al., 2002; Li et al., 2002) which may change bacterial location and reduction of planktonic cells. Additionally, unlike our study Colehour et al. (2014) found predominance of L. reuteri which is known to antagonize S. salivarius (Nikawa et al., 2004). Similar to previous studies (Colehour et al., 2014; Elizaquível et al., 2015; Puerari, Magalhães-Guedes & Schwan, 2015), Lactobacillus was a dominant genus of lactic bacteria in chicha found in both culture dependent and independent assessments.

Our study complements previous microbiological analyses carried out in chicha and shows for the first time the potential adaptation of S. salivarius, S. mutants (and possibly other streptococci from the human upper respiratory tract) to grow in cassava mush. The study not only shows how bacteria from human microbiota may adapt to artisanal fermentative processes but also shows that chewed chicha may potentially transmit human pathogens such as S. mutans, one of the causative agents of dental plaque and cavities (Loesche, 1986); Streptococcus mutans can be transmitted person-to-person, most likely through saliva (Baca et al., 2012). This is especially relevant because these types of beers are consumed as early as two or three days after preparation.

The main limitation of our study was the low number of samples analyzed of each beer. However this limitation does not invalidate the main findings of this study. Additionally, the culture medium (MRS) is not suitable to culture Lactobacillus from cereals (Minervini et al., 2012), therefore we may have underestimated the bacterial diversity in these beers.

We thank Mariela Serrano for her technical advice and Danny Navarrete for the statistical analysis.

Additional Information and Declarations

Competing Interests

Author Contributions

DNA Deposition

Data Deposition

The authors declare that they have no competing interests.

Ana L. Freire performed the experiments, analyzed the data, wrote the paper, prepared figures and/or tables, reviewed drafts of the paper.

Sonia Zapata conceived and designed the experiments, wrote the paper, reviewed drafts of the paper.

Juan Mosquera performed the experiments, analyzed the data, prepared figures and/or tables, reviewed drafts of the paper.

Maria Lorena Mejia analyzed the data, reviewed drafts of the paper.

Gabriel Trueba conceived and designed the experiments, wrote the paper, reviewed drafts of the paper.

The following information was supplied regarding the deposition of DNA sequences:

Genbank, accession numbers from KT722809–KT722833.

SRA SRP070493; SRP070493, SRS1299611, SRX1612367, SRR3202831, SRS1299612, SRX1612366, SRR3202830, SRS1299613, SRX1612365, SRR3202829, SRS1310202, SRX1600290, SRR3187397, SRS1310203, SRX1600289, SRR3187396, SRS1310204, SRX1600288, SRR3187395, SRS1310207, SRX1612364, SRR3202828, SRS1310208, SRX1600292, SRR3202832.

The following information was supplied regarding data availability:

The research in this article did not generate any raw data.

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
