# Peer review of "Bacteria associated with human saliva are major microbial components of Ecuadorian indigenous beers (chicha)"

_PeerJ, doi:10.7717/peerj.1962_

## Round 0.1 · original submission · Major Revisions

· Academic Editor

Major Revisions

Both expert reviewers provided extensive suggestions for further improvement, regarding missing information, experimental design and methods, as well as presentation and interpretation of data.

Reviewer 1 ·

Basic reporting

The article describes the culture dependent and culture independent analysis of Ecuadorian indigenous beers. Different from alcoholic fermentations in Europe and Asia, where starch hydrolysis is achieved by malt and fungal amylases, Ecuadorian fermentations are based on saliva amylase. The manuscript provides one of few detailed analyses of fermentation microbiota of Chica and may thus make a relevant addition to the experimental literature

Experimental design

The experimental design is generally valid but has two shortcomings.
- If 16S rRNA sequences are used for taxonomic identification of isolates, the query sequences should be compared to the sequence of type strains, not to genebank sequences. Several databases e.g. the ribosomal database project provide this comparison (http://rdp.cme.msu.edu/)
- MRS is not a suitable medium for growth of cereal-associated lactobacilli (see e.g. Appl. Environ. Microbiol. 78 1251-1264). Accordingly, many of the lactobacilli are not cultivated on MRS. This shortcoming can't be fixed without additonal sampling; authors should comment that their culturing methods are not sufficient for full recovery of all fermentation organisms.

Validity of the findings

Documentation of the high throughput sequencing is insufficent. The manuscript should describe which primers were used to generate the amplcions, the minimum length of the sequences, how many sequences were obtained (per samples), how sequences were processed, and how sequences were assigned to bacterial species. Moreover, an accession number to the sequence data should be provided.

Additional comments

W. confusa, not W. confuse...

Overall, a nice piece of work.

·

Basic reporting

The paper reports the bacterial composition of indigenous beers of Ecuador, which is interesting and relevant to the understanding of traditional fermentation practices and the microbiology of foods in general. Overall the paper is clear and well-written, but there are some small typographic and grammatical errors throughout. Before publication, make sure to thoroughly copy edit. Here are some errors I noticed, though this list is not complete:

Line 15:
-I think you mean “rely” instead of “relay”

Line 27:
-should be “associated with” not “associated to”

Line 30:
-“may have reduced”

Line 43:
-more clear to say “culture dependent and culture independent techniques”

Line 50:
-should be “ground” corn

Line 105:
-Do you mean Weissella confusa?

Line 146:
-Do you mean “chicha” rather than "chichi"? I don't know what the plural of chicha is so this may be intentional, but I know that my computer autocorrected chicha to chichi a few times so perhaps this is an error!

For the figures, this is a minor point, but I think making the x-axis simply be "time" with 0, 24, 48, and 72 hours marked would make more sense than having numbers 1-4 and the time written in parenthesis underneath.

It's not strictly necessary but I think the data in table 3 could be presented in ways that are more useful to the reader. Perhaps even listing the species in order of prevalence in the different beers would be useful for understanding how the beers cluster. Data of this type is also frequently presented with the species in a phylogenetic tree. Perhaps that would be useful in this case too.

From looking at table 3 it seems like the two beers made from cassava are more similar to each other than those made with the other starting material. It would be interesting to give some statistical measure of similarity for those two compared to the others, or perhaps to visualize it through some kind of factor analysis. Perhaps this would become more relevant if you were to expand your analysis with more samples of each type.

Experimental design

The methods are straightforward and appropriate for the study and the conclusions.

Validity of the findings

I agree with your final statement that you are limited by the small number of beers that have been sampled. Would you speculate that all other cassava-based beers may have oral-associated bacteria regardless of whether they are chewed or not? Or is this more an indictment of the particular producer and their hygiene? Perhaps giving the reader some more context about the practice of chewing the cassava—is there more anthropological data as to why it would be chewed vs. mashed and how common one is compared to the other? Future experiments looking at a broader set of beer samples would be very interesting, as would be a time-course during the fermentation process.

Additional comments

I think it would be useful to include a very short discussion about the yeasts that are present in the chicha and why you chose to focus on the bacteria. Without mentioning the yeast it seems like a very big gap in understanding the microbiology of the beer!

Also, in the discussion you mention previous studies that don’t find oral-associated bacteria in chicha. I think this is worth mentioning in the introduction in order to give more context to your motivation for doing the study and to the results that you are finding.

Another issue in terms of context for the discussion is understanding the timing of how the beer is produced and consumed. At what point in the fermentation process is the beer usually ready for drinking? That seems relevant to the discussion of 4-day vs. 3-day fermentation and the presence of oral-associated bacteria. If the beer is typically consumed after only 3 days, then your comment about the transmission of potential pathogens is particularly relevant. Otherwise, if the practice is to wait longer until the lactobacillus have outcompeted the strep bacteria that would also be quite notable.

---

## Round 0.2 · Minor Revisions

· Academic Editor

Minor Revisions

Your manuscript has much improved, and there are only a few clarifying questions remaining.

Also please make sure to carefully proofread your manuscript before resubmission.

Reviewer 1 ·

Basic reporting

see below

Experimental design

see below

Validity of the findings

see below

Additional comments

Comments to the first review were addressed appropriately. The culture dependent analysis remains a bit wonky but the shortcomings do not question the main conclusion, i.e. presence of oral streptococci.

Remaining comments for improvement are minor:

- There is a bit of inconsistency with respect to the number of isolates per sample. Based on the materials and methods section, 40 to 80 colonies in total were isolates and 24 to 40 (total) were sequenced? That should provide a higher number than the 24 that are listed in the results section.

Also, the materials and methods section states "Ten to twenty colonies (showing different morphology) were randomly picked from each sample. Six to ten colonies that had the characteristics of lactic acid bacteria (oxidase negative, catalase negative...)"

Which group of organisms makes up the difference between "ten to twendy" and "six to ten"? If that is the yeast count, please specify.

·

Basic reporting

The authors addressed all my concerns and have done a good job with the final version of the paper. There are only a few very minor copy editing issues introduced by the new edits, mostly incorrect spacing between words. Just make sure to catch those before the final version!

Experimental design

No comments

Validity of the findings

It was great to see the principal component analysis and the fungal species, thank you for including that.

---

## Round 0.3 · accepted · Accept

· Academic Editor

Accept

There are still several minor issues with the text, which you should check carefully. For example, I spotted the following items:

l.16: relies, rather than rely
l.16: fermented substrates, rather than fermenting substrates
l.19: 16S ribosomal RNA (rRNA) gene, rather than just 16S. This should also be adjusted throughout.